# Scale-Up of Mixing Equipment for Suspensions

**Tomáš Jirout \***[ID]**, František Rieger and Dorin Ceres**

Department of Process Engineering, Faculty of Mechanical Engineering, Czech Technical University in Prague, Technická 4, 166 07 Prague 6, Czech Republic; Frantisek.Rieger@fs.cvut.cz (F.R.); DorinC@seznam.cz (D.C.)

\* Correspondence: Tomas.Jirout@fs.cvut.cz; Tel.: +420–224–352–681

**Abstract:** This paper deals with the scale-up of equipment for the mixing of suspensions. The measurement of just-suspended agitator speeds was carried out with standard, pitched, four-blade turbines and folded, four-blade turbines in three vessels (290 mm, 600 mm, and 800 mm in diameter) for several particle sizes and concentrations. The results of measurements confirmed that scale-up based on dimensionless Froude number dependence, on the relative particle size and concentration, can be used. On the basis of the results, a scale-up rule for agitator speeds in a given suspension and equipment geometry was recommended, and various conclusions reported by different investigators were discussed.

**Keywords:** mixing; axial agitator; suspension; scale-up

---

## 1. Introduction

Mixing suspensions is a very frequent operation in process industries. Most mixing experiments are carried out at the laboratory scale using geometrically similar models. In order to use the results for the design of industrial mixing equipment, knowledge of scale-up rules is necessary. Power consumption per unit volume is often recommended as a scale-up rule for suspension-mixing equipment. However, studies of the relationship of power per unit volume by other authors, presented in [1,2], indicate that extremely various conclusions have been reported by different investigators; e.g., specific power scale-up exponent δ (Equation (6)) listed in the overview in the publication [1] reached values from −1 up to 0.5. The aim of this publication is to clarify this discrepancy using an objective experimental method for identification of the just-suspended state in a wide range of sizes and concentrations of particles in a mixing apparatus equipped by two different types of axial flow impellers.

Our first experimental results on this topic were presented in [3]. The results of our more systematic research on this topic were reported in [4,5]. The experiments presented in [4,5] were carried out with pitched, six-blade turbines and volumetric glass ballotine content up to 10%. This paper presents the results of scale-up experiments for a larger number of types of impellers (pitched, four-blade turbine and diagonally folded, four-blade turbine) and especially for a wider range of volumetric concentration of particles up to 20%. The described scale-up procedure can be used for direct designing of industrial agitated tanks, rectors, and biorectors based on data from model experiments.

## 2. Theoretical Background

The following dimensionless suspension characteristic was suggested in [6] for evaluating critical (just-suspended) agitator speed measurements under a turbulent regime for a given concentration of the solid phase and type, including geometrical arrangement of the agitator in the vessel.

$$\mathrm{Fr}' = f(d_P/D) \tag{1}$$

where $d_p$ is particle diameter (mm), D vessel diameter (mm), and Fr′ = $(n^2d\rho)/(g\Delta\rho)$ [-] modified Froude number (n is just-suspended agitator speed ($s^{-1}$), d is agitator diameter (mm), g is gravity acceleration ($m/s^2$), ρ is liquid density ($kg/m^3$), and Δρ is solid/liquid density difference ($kg/m^3$)).

For relatively small particles, the dependence between the modified Froude number and a dimensionless particle size can be formulated in the power form.

$$\text{Fr}' = C(d_p/D)^\gamma \tag{2}$$

where γ is the exponent.

The values of coefficient C and exponent γ depend on particle volumetric concentration c [-]. A mathematical description of these dependencies was proposed in [7] in the form of:

$$C = A \exp(Bc) \tag{3}$$

$$\gamma = \alpha + \beta c \tag{4}$$

where A, B, α and β are constants. The cvalue of these constants depends only on a geometrical configuration of mixing equipment (vessel and agitator shape and geometry).

From the definition of Fr′ and Equation (2), we get the following scale-up rule for the agitator speed in a given suspension and equipment geometry:

$$n\,D^\kappa = nD^{(1+\gamma)/2} = \text{constant} \tag{5}$$

where κ is the agitator speed scale-up exponent.

From the definition of the power number Po (Po = $P/(\rho n^3 d^5$ [-]), the following scale-up rule for specific power in turbulent region can be obtained:

$$P/V \sim n^3D^2 \sim D^{(1-3\gamma)/2} = D^\delta \tag{6}$$

where P is power (W) and δ is specific power scale-up exponent.

From the conclusions presented in [6], it follows that, at small, sub-critical values, $d_p/D$ is exponent γ positive and Fr′ increases with $d_p/D$; at higher, super-critical values, $d_p/D$ is γ approximately 0. This is illustrated in the results of visual experiments with a pitched, four-blade turbine in a vessel with a dish bottom, presented in [8], which are depicted in Figure 1. However, equipment of industrial size usually operates in the sub-critical region ($d_p/D < 0.005$).

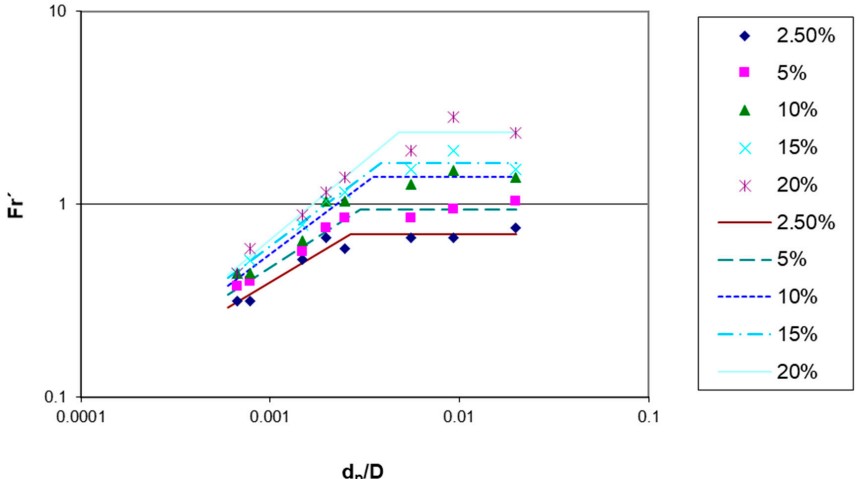

**Figure 1.** Dependence of Fr′ on relative particle size $d_p/D$ and mean volumetric concentration of particles c, presented in [8] (points—experimental results, line—regression according to Equation (2) by least-squares regression analysis).

## 3. Experimental Section

The model and pilot-plant-scale experiments were carried out with two types of axial-flow impellers with four blades (see Figure 2). The pitched, four-blade turbine with pitch angle 45° was selected to represent standard types of impellers commonly used in an industry. Due to the generalization of the results representative of the hydrodynamically optimized (hydrofoil) impeller, the diagonally folded-blade impeller was tested. Impeller diameter d was equal to one-third of the diameter of the vessel D, i.e., D/d = 3. The impeller off-bottom clearance measured from the lowest point on the blades $H_2$ was chosen near optimum from the point of view of operating conditions and minimal power consumption necessary for particle suspension (e.g., [9]). The impeller off-bottom clearance $H_2$ of the pitched, four-blade turbine was equal to 0.5 d and for the diagonally folded, four-blade turbine was equal to 0.75 d. Both impellers were operated to pump liquid down toward the bottom of the vessel. The scale-up rule was tested by measurement in geometrically similar configurations of flat-bottomed vessels with 290 mm, 600 mm, and 800 mm diameters. The height of the liquid level, H, was equal to the inner diameter of the vessel, D. All experimental vessels were equipped with four standard radial baffles for elimination of the formation of the central vortex with width b equal to 0.1 D. The arrangement of the experiment is shown in Figure 3.

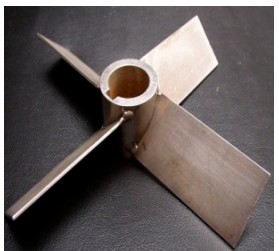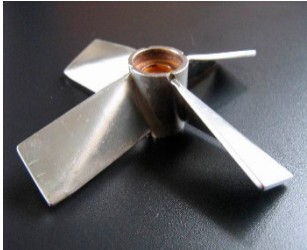

**Figure 2.** Tested agitators.

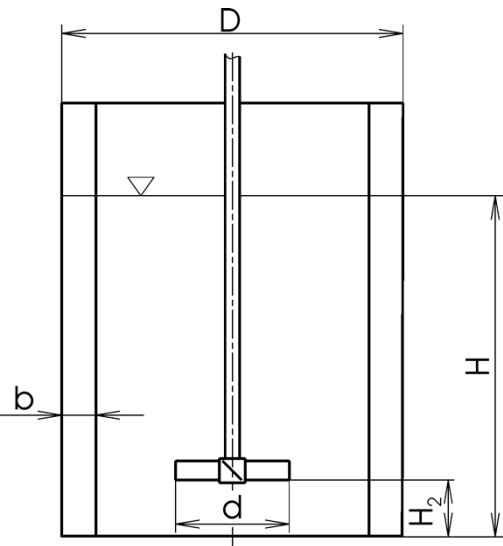

**Figure 3.** Geometrical arrangement of the experimental mixing equipment (geometrical parameters of vessel: D = 290, 600, and 800 mm; H/D = 1; b/D = 0.1).

The agitated batch for all model and pilot-plant experiments was composed by monodisperse suspensions of spherical glass particles in 2.5 b.w.% water solutions of sodium chloride. The diameters of the monodisperse particles varied in a range from 0.18 mm to 0.9 mm. Mean volumetric concentration of particles was in a range from 2.5% to 20%. The just-suspended agitator speeds were measured by an

advanced experimental method designed by leading authors of this article based on application of an electrochemical method described, e.g., in [10], and checked visually.

The dependence of the arithmetic mean value and the standard deviation of the probe electrodiffusion (ED) current on the setting agitator speed was used for identification of the just-suspended state. A typical form of this dependence is shown in Figure 4. In suspension measurements, it is possible to observe three typical states of the behavior of the particles in the suspension near the ED probe. This behavior of the particles changes with increasing of the agitator speed. First, a particle settled on the vessel bottom and covered all surface of the ED probe (Figure 4—state A). The particle layer became thinner as the agitator speed increased. After that, the particles were alternately suspended and settled on the vessel bottom (Figure 4—state B). After reaching just-suspended state, the suspension flowed fully along the ED probe; i.e., no particles settled motionless on the vessel bottom (Figure 4—state C). These three suspension states can be observed by an electrochemical method. The just-off-bottom particle suspension state is represented by the practically jump increase in the measured ED current or its deviation (see Figure 4). ED current was measured in the dependence of the impeller speed during all suspension experiments. These data were recorded and the just-suspended impeller speed was automatically evaluated.

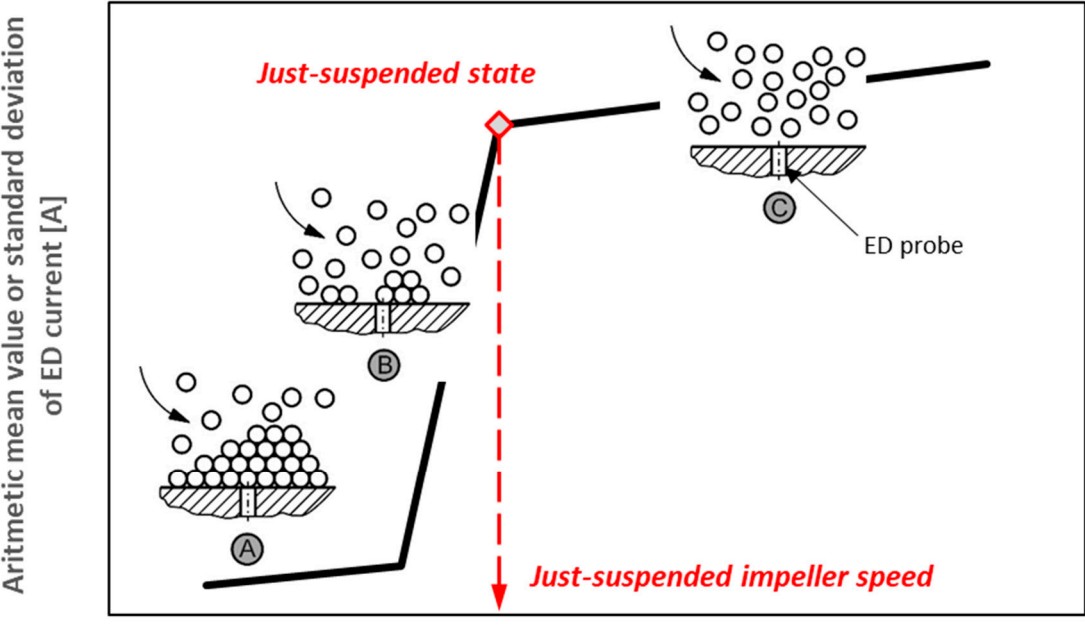

**Figure 4.** Typical dependence of the measured electrodiffusion (ED) probe electric current on the agitator speed with the representation of the typical states of particles behavior in the suspension at the vessel bottom [10].

For use of the electrochemical method in experiments, the vessels of laboratory and pilot-plant mixing equipment were equipped with six working electrodiffusion (ED) probes made from platinum wire with diameter 0.5 mm flush mounted at the vessel bottom in the center and nearby vessel wall (see Figure 5).

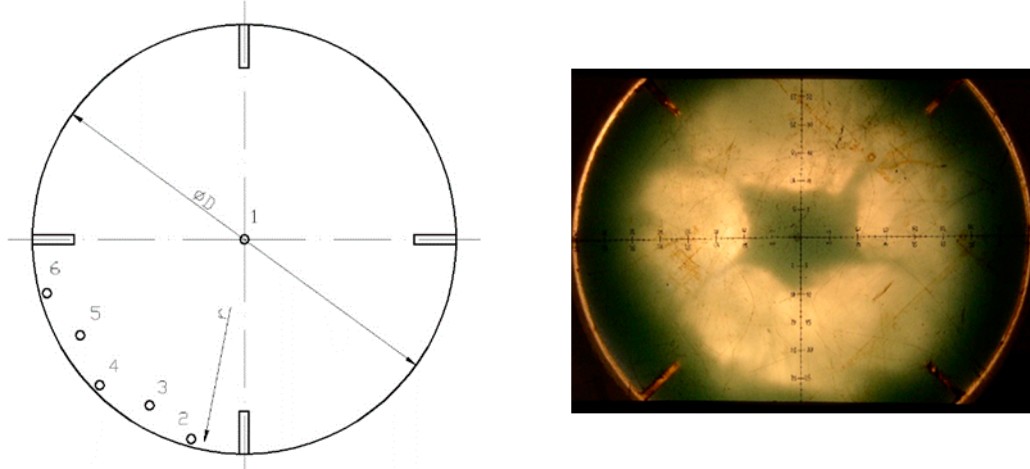

**Figure 5.** Location of electrodiffusion probes mounted at the vessel bottom (probe 1 – 2r/D = 0; probe 3, 5 – 2r/D = 0.89; probe 2, 4, 6 – 2r/D = 0.96).

## 4. Experimental Results

Experimental data were evaluated in the form of dimensionless suspension characteristics according to Equations (2)–(4) for all tested scales together by the least-squares regression method. Non-linear regression function "nlinfit" from the MATLAB system was used. The confidence intervals of the constants A, α, and β were estimated max. 15% and for the exponent B max. 10%. The achieved accuracy of regression evaluation was better than it was achieved in publications [6,7], in which the regression form of dimensionless characteristics was proposed and verified.

The verification of the scale-up rule is illustrated in Figures 6–8, which present the results of measurements with pitched, four-blade turbines and folded, four-blade turbines at different concentrations. In Figures 6 and 7, the results for pitched, four-blade turbines in suspensions with concentrations c = 0.05 and 0.1 are presented. In Figure 8, the results for folded, four-blade turbines in suspensions with concentrations c = 0.075 are presented. No significant differences between the results for vessels of different sizes can be observed in these figures. Similar results were obtained for both agitators at different particle contents. This confirms that scale-up based on Fr'= f (d$_p$/D, c) relations can be recommended.

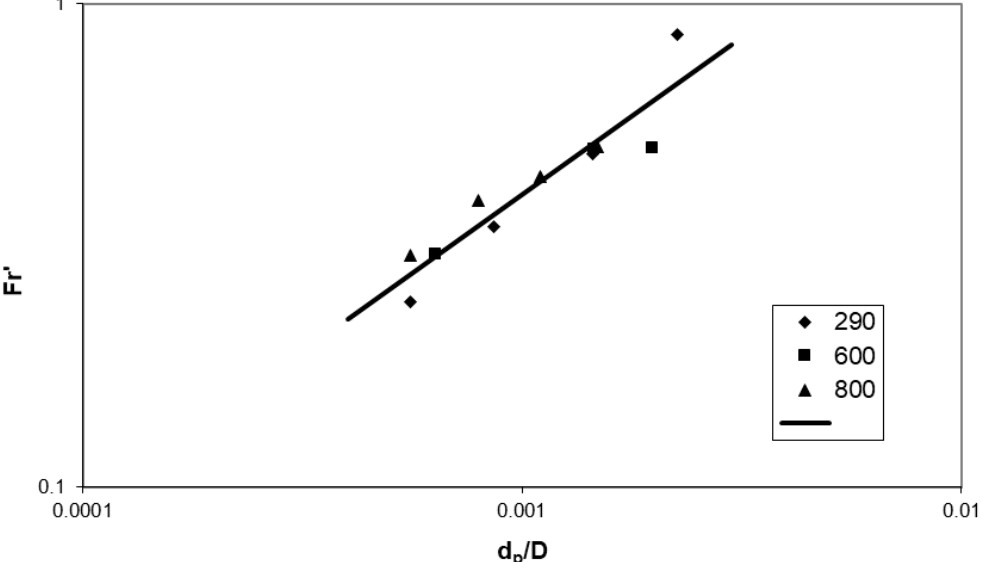

**Figure 6.** Dependence of Fr' on relative particle size for pitched, four-blade turbines for mean volumetric concentration of particle (5%).

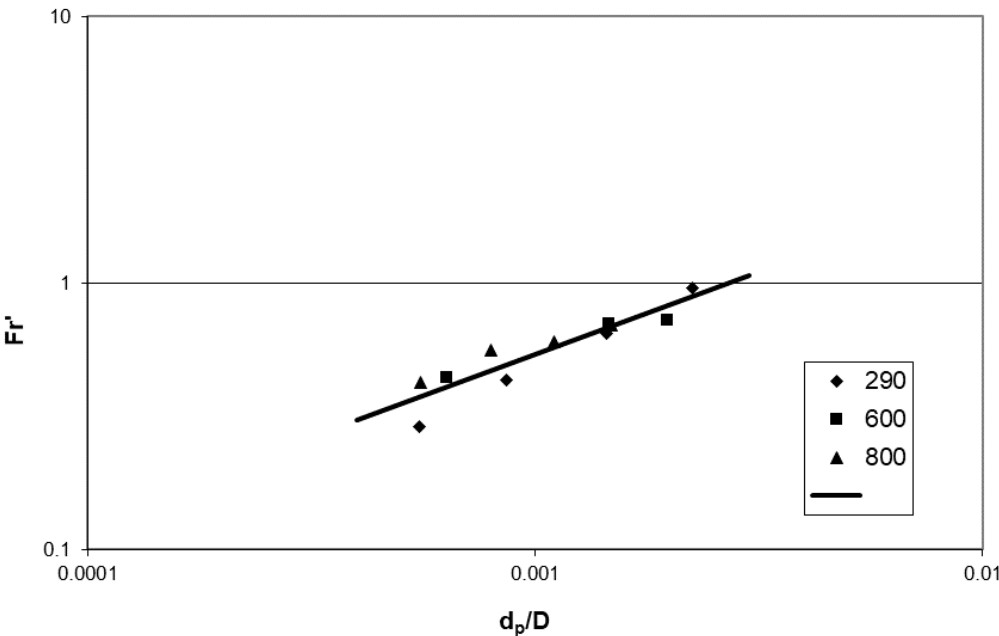

**Figure 7.** Dependence of Fr′ on relative particle size for pitched, four-blade turbines for mean volumetric concentration of particle 10%.

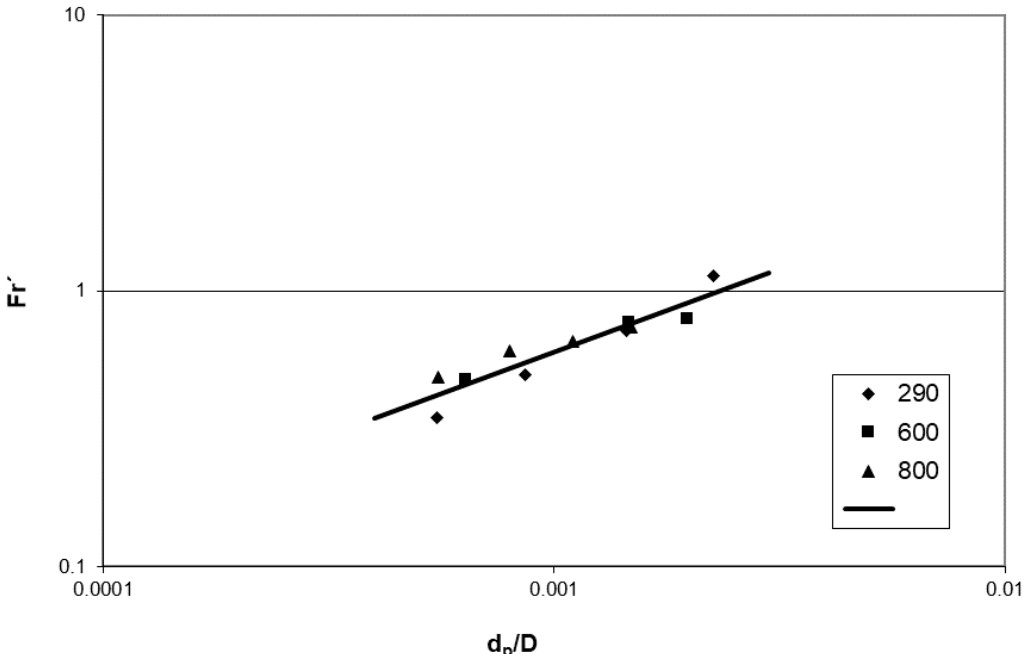

**Figure 8.** Dependence of Fr′ on relative particle size for pitched, four-blade turbines for mean volumetric concentration of particle (7.5%).

From previous Figures 6–8 in logarithmic coordinates, it is obvious that the power form of data evaluation by Equation (2) is acceptable. The dependence of exponent $\gamma$ and coefficient C on particle volumetric concentration c are shown in Figures 9–12. In accordance with Equations (4) and (5), the exponent $\gamma$ grows linearly with increasing volumetric particle concentration c, and the dependences of coefficient C on the particle concentration c has exponential form; i.e., in semi-logarithmic coordinates, it can be approximated by straight lines.

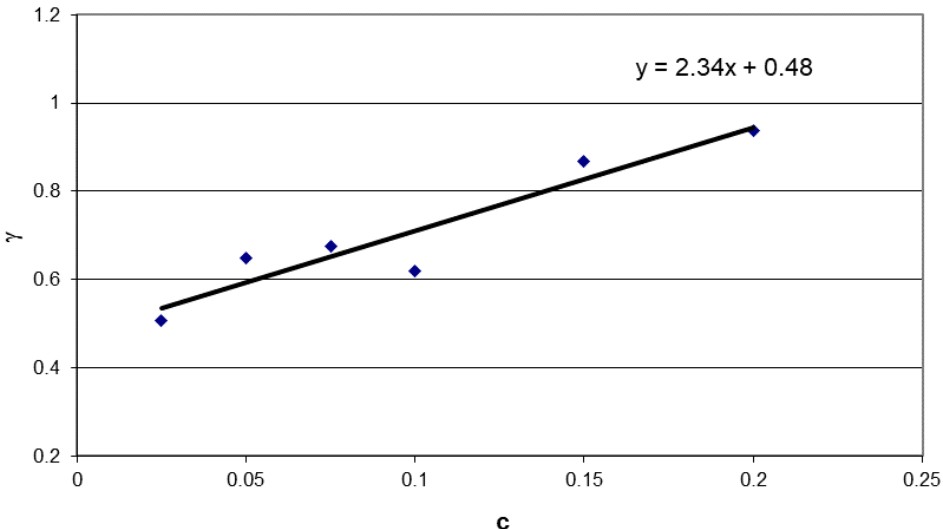

**Figure 9.** Plot of exponent γ on c for pitched-blade turbine.

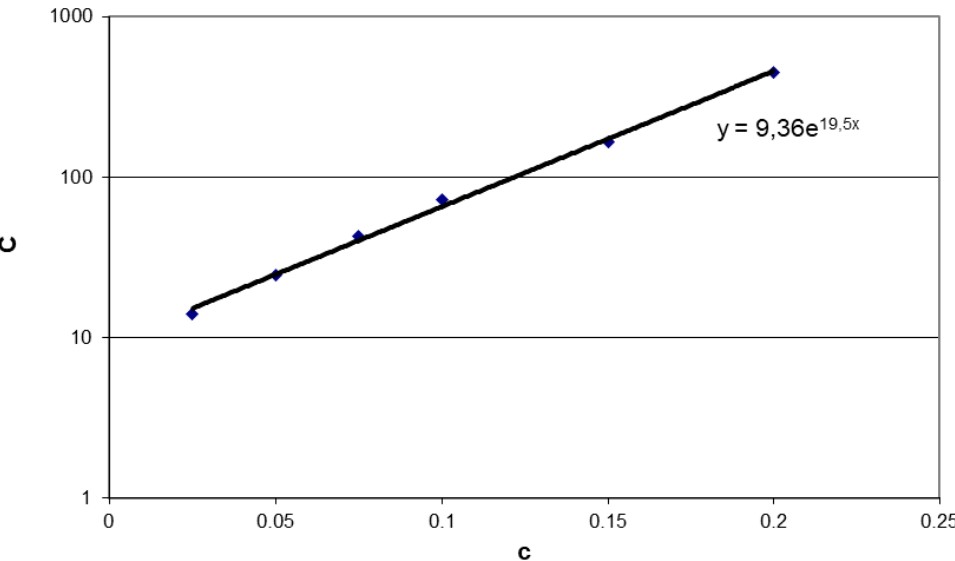

**Figure 10.** Plot of coefficient C on c for pitched-blade turbine.

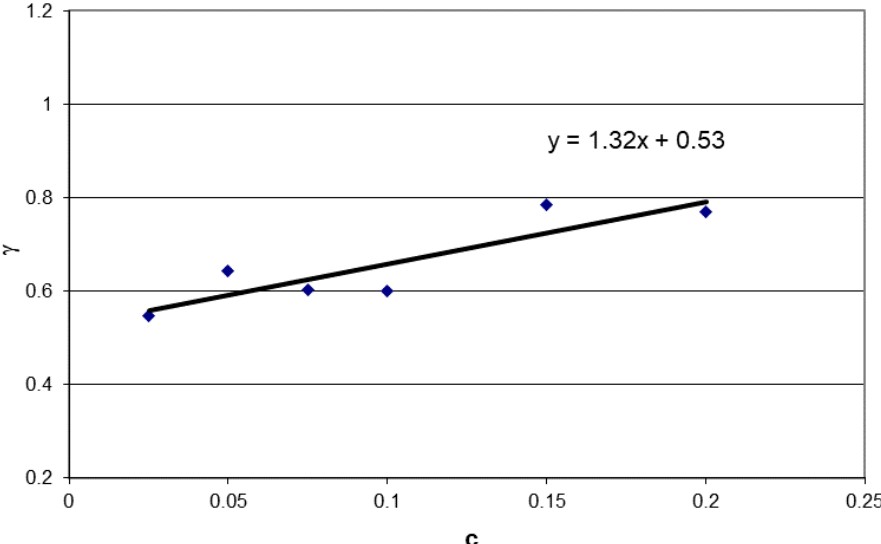

**Figure 11.** Plot of exponent γ on c for folded, four-blade turbine.

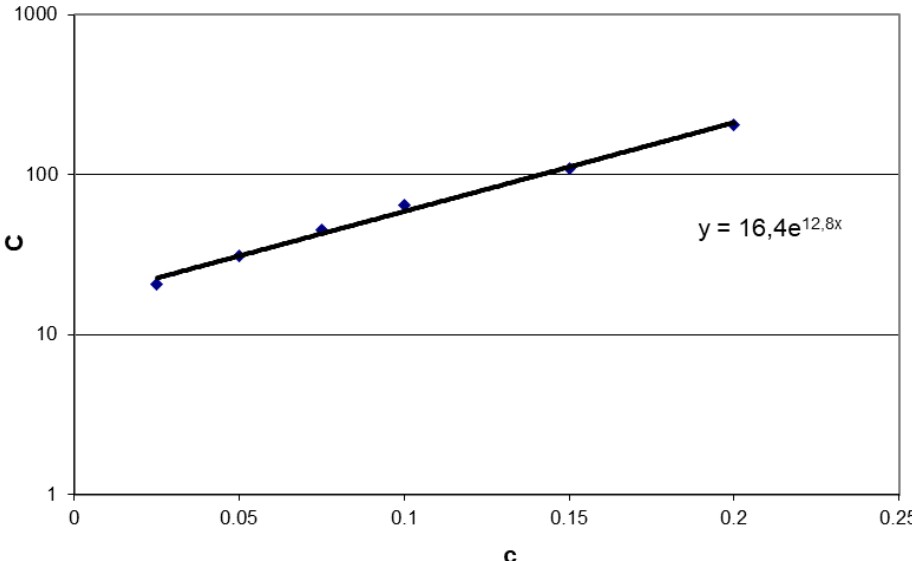

**Figure 12.** Plot of coefficient C on c for folded, four-blade turbine.

## 5. Discussion

Dependences of exponents $\gamma$, $\kappa$ and $\delta$ on concentration c for both agitators are shown in Figures 13–15. In Figures 14 and 15, mean value of exponents for both agitators is also depicted.

Figure 15 shows that scale-up at constant specific power is on the safe side at all concentrations. Figure 14 shows that exponent $\kappa$ tends to 1, which means scale-up at constant tip speed and specific power inversely proportional to equipment size can be recommended for high concentrations. It means that mixing in large volumes, especially for concentrated suspensions, brings significant savings of energy.

At super-critical values of $d_P/D$, $\gamma = 0$, $\delta = 1/2$ (see Figure 1), and specific power increases with equipment size, in which case we can meet up in suspensions of relatively large particles in small vessels of laboratory size.

Zwietering, in his classical pioneering paper [11], recommended constant value $\delta = -0.55$, which corresponds with particle concentration 10–13% approximately (see Figure 15). At higher concentrations is Zwietering's recommendation on the safe side. At smaller concentrations, operation of equipment at a speed smaller than critical with a thin particle layer on the vessel bottom can be acceptable.

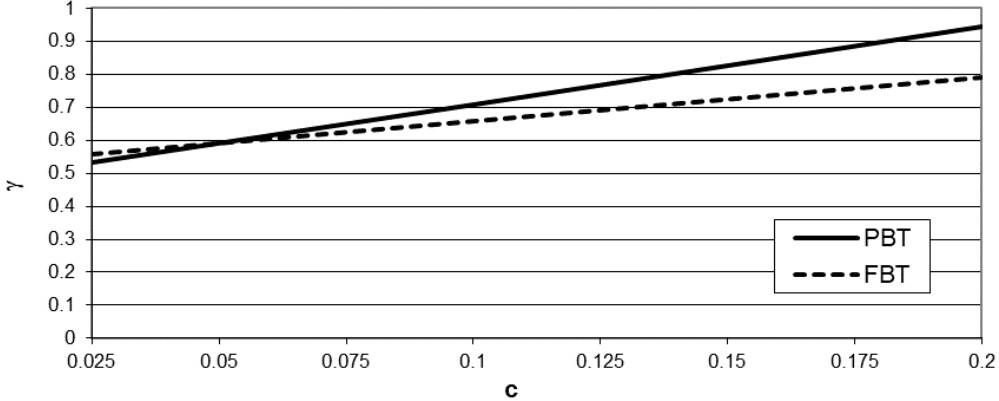

**Figure 13.** Dependence of $\gamma$ on c for both agitators.

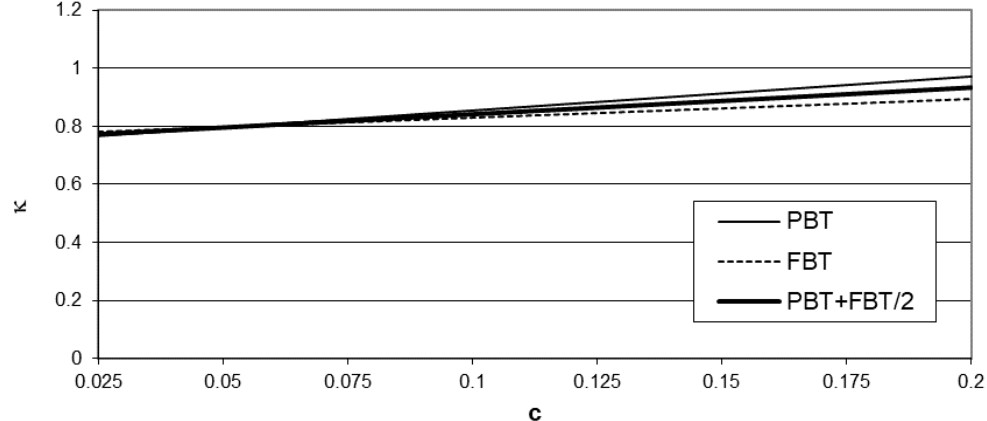

**Figure 14.** Dependence of κ on c for both agitators.

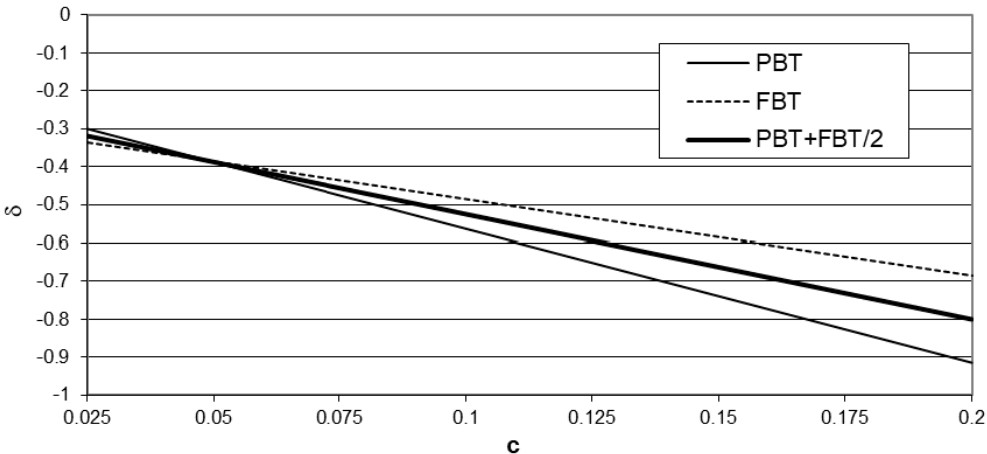

**Figure 15.** Dependence of δ on c for both agitators.

**Author Contributions:** Conceptualization, T.J. and F.R.; methodology, T.J. and F.R.; design of experiments, T.J.; experiments, D.C.; data analysis, T.J.; writing—original draft preparation, F.R and T.J.; writing—review and editing, T.J. All authors have read and agreed to the published version of the manuscript.

**Funding:** This research was funded by the Technology Agency of the Czech Republic, grant number TA02010243, "Mixing Equipment for Sludge Processing", and grant number TA02011251, "Optimization of enameled mixing equipment according to technological needs of end users".

**Conflicts of Interest:** The authors declare no conflict of interest.

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
