# Peer review of "Scale-Up of Mixing Equipment for Suspensions"

_processes, doi:10.3390/pr8080909_

Round 1

Reviewer 1 Report

Dear Authors

I am not a native speaker but check lines 31, 44, 65 at least. They do look strange to me. 

I hope I got it right. You try to determine the mixing speed to achieve the just-suspended state and then use the n in the Po to determine the power. Some additional explanation in the Introduction might be useful. 

Regarding your vessels: were the scaled up in diameter only or in height too

line 73 did you use several particle classes (according to diameter) or one mixture of different diameters. 

Figure 2 Is the line some kind of approximation of discrete values? Explain

why duplicate legends.

Since difference in density is an important factor. Did you control the temperature of the suspension or did you measure actual density. Did you experience any issues with air bubbles? 

Best regards

Reviewer 2 Report

Review of the paper (Ref. processes-86188)

“SCALE-UP OF MIXING EQUIPMENT FOR SUSPENSIONS”

by Tomáš Jirout *, František Rieger and Dorin Ceres

           This paper presents research into dispersion of solid particles in liquid with the use of different scale mixing vessels. The Authors investigated such a process for two various, high-speed impellers (standard, pitched four-blade turbine and folded four-blade turbine) in a relatively wide range of particle concentrations and impeller speeds.          

The paper represents a lot of work and results are presented in an interesting manner, using physical modeling first of all.

            However, I have several comments that the Authors could consider when revising the manuscript: 

1)     In my opinion the Authors should give a brief comment on how their results can be used in other, specific (examples) chemical and engineering systems, along with any disadvantages and advantages.

2)     The Authors should describe in more detail investigated mixing vessel, including information on its geometry, e.g. liquid height in the vessel, width and arrangement of baffles, open or closed vessel etc.  I also think it would be beneficial to explain in this paper why impeller clearances were different for both impellers.

3)     Also the Authors should describe in more detail the regression method used for determining values of constants, exponents and coefficients, provide the accuracy of their estimation, errors and/or confidence intervals.

In summary, I think this paper can be published in Processes after revision (with small changes), taking above-mentioned remarks into consideration. 

Round 2

Reviewer 1 Report

The authors have considered my remarks and made some additional changes. I believe the paper has improved. I recommend some work on English language (but I am not a native speaker).